# Rare Earth Elements and Sr Isotope Ratios of Large Apatite Crystals in Ghareh Bagh Mica Mine, NW Iran: Tracing for Petrogenesis and Mineralization

**Narges Daneshvar** [1],*[ID], **Hossein Azizi** [1],*[ID], **Yoshihiro Asahara** [2][ID], **Motohiro Tsuboi** [3] **and Mahdi Hosseini** [4]

1 Department of Mining Engineering, Faculty of Engineering, University of Kurdistan, Sanandaj 66177-15175, Iran
2 Department of Earth and Environmental Sciences, Graduate School of Environmental Studies, Nagoya University, Nagoya 464-8601, Japan; asahara@eps.nagoya-u.ac.jp
3 Department of Applied Chemistry for Environment, School of Science and Technology, Kwasei Gakuin University, Sanda 669-1337, Japan; tsuboimot@kwansei.ac.jp
4 Department of Geology, Faculty of Science, Imam Khomeini International University, Qazvin 34148-98818, Iran; Hosseini_qazvin@yahoo.com
* Correspondence: daneshvar_n@yahoo.com (N.D.); azizi1345@gmail.com (H.A.); Tel.: +98-918-872-3794 (H.A.)

**Abstract:** The 320 Ma Ghareh Bagh mica mine is the only active mica mine in northwest Iran, and hosts Mg-bearing biotite (phlogopite) with apatite, epidote, and calcite. Chemical investigation of apatite infers the high abundances of the rare earth elements (REEs up to 5619 ppm), higher ratios of the LREE/HREE $((La/Yb)_N = 28.5–36.7))$ and high content of Y (236–497 ppm). REE pattern in the apatite and host A-type granite is almost the same. Ghareh Bagh apatite formed from the early magmatic-hydrothermal exsolved fluids at the high temperature from the Ghushchi alkali feldspar granite. The apatite crystals came up as suspension grains and precipitated in the brecciated zone. The early magmatic-hydrothermal fluids settle phlogopite, epidote, chlorite, K-feldspar and albite down in the brecciation zone. Due to the precipitation of these minerals, the late-stage fluids with low contents of $Na^+$, $Ca^{2+}$ and REE affected the early stage of alteration minerals. The high ratios of $^{87}Sr/^{86}Sr$ (0.70917 to 0.70950) are more consistent with crustal sources for the apatite large crystals. The same ages (320 Ma) for both brecciated mica veins and host alkali feldspar granites infer the apatite and paragenesis minerals were related to host granite A-type granite in the Ghareh Bagh area.

**Keywords:** apatite; rare earth elements; Sr isotope; phlogopite; Ghareh Bagh mica mine; Iran

## 1. Introduction

Apatite, with the general formula of $Ca_5(PO_4)_3(F, Cl, OH)$, is a well-known mineral group as calcium phosphate. Apatite is a key mineral candidate for recording the trace elements chemistry of the host rock as its crystallization time, because of the following reasons: (1) It is an early crystallizing mineral and its stable for a long time during the evolution of various silicate melts [1]. (2) Apatite is a very common mineral in the various types of rocks such as igneous, metamorphic, and sedimentary rocks, as well as hydrothermal systems [2]. (3) Chemistry of apatite is a function of the crystallization environment at the time of apatite crystallization [2–4]. (4) Its resistance to physicochemical weathering and is stable during diagenesis and sediment transport, and so it can preserve the original geochemical signature [5–8]. A large range of contents of minor and trace elements such as rare earth elements (REEs), Sr, Th, U and Y, can structurally accommodate in apatite with substituting on cation and anion sites [2]. $REE^{3+}$ and $Y^{3+}$ can enter the apatite's structure by the some substitutions [9]: (1) $2REE^{3+}$ ($Y^{3+}$)

+ [V] = 3Ca$^{2+}$; (2) REE$^{3+}$(Y$^{3+}$) + Na$^+$ = 2Ca$^{2+}$, and (3) REE$^{3+}$ + Si$^{4+}$ = Ca$^{2+}$ + P$^{5+}$, where [V] denotes the vacancy. The composition of the source rocks can control REE patterns of apatite [10]. Furthermore, REE behavior in apatite is very sensitive to the variation of some parameters such as pH and redox condition, temperature, and fluid composition [11,12], and also, it is sensitive to co-crystallizing minerals such as plagioclase [13]. Therefore, based on the geochemical composition of apatite, lots of robust information about geochemistry, petrogenesis of parental magma, and evolution of magma could be yielded, and it has been widely used to obtain substantial genetic information [2,9,14]. The low solubility of P$_2$O$_5$ in magma causes apatite to crystalize as an early magmatic phase, but it may continue to crystallize from the hydrothermal fluid and metasomatism [15,16]. Subsequently, the composition of apatite can play a fundamental role for mineral exploration, especially in the magmatic-hydrothermal deposit. In addition, apatite is too useful mineral to study the geochemical history of a magmatic-hydrothermal system [3,14,17].

Moreover, apatite is an important Sr- enriched mineral and is known for lack of Rb, and thus the value of Rb/Sr is very low in this mineral. Due to the lack of Rb, the $^{87}$Sr/$^{86}$Sr ratio of apatite indicates the initial $^{87}$Sr/$^{86}$Sr ratio of the sample [18–20]. Two factors—rapid cooling and no thermal disturbances—are necessary for the preservation of $^{87}$Sr/$^{86}$Sr in apatite [21]. Several studies have shown that the Sr isotope ratio of apatite is almost not significantly was affected by post-crystallization events such as hydrothermal alteration [18–20]. As a result, the isotopic value of apatite makes it a prevailing tool to trace the source of magma [19,20].

Some apatite-bearing ore deposits are well-reported in Iran, including iron oxide apatite ore in central Iran like Ce-Chahoun, Esfordi and Lakeh Siah [17,22,23], and magnetite–apatite with hydrothermal source in south Zanjan including Zaker, Morvarid, Sorkheh-Dizaj and Aliabad [24]. Herein, we focus on the genesis of the Ghareh Bagh apatite-bearing phlogopite deposit in NW Iran. In the Ghareh Bagh mica deposit, phlogopite production started in 1969 [25] with a production of over 800 tons per year with a reserve of approximately 65,000 tones [26].

The mineral assemblage in Ghareh Bagh deposit comprises phlogopite, apatite, epidote, chlorite, and calcite; of those, phlogopite has economic importance and it is actively mined now. The present study contains the first detailed geochemical data (REEs) and Sr isotope ratios on the apatite grains in the Ghareh Bagh area. The goal of this study is to identify the source of apatite, the genetic relationship between apatite and the host rocks. Furthermore, physicochemical condition of the parental magma to monitor the magmatic-hydrothermal process which has played a key role for the genesis of apatite large crystal.

## 2. Geological Setting and Petrographic Studies

The Ghareh Bagh mica mine is situated in the north of Urmia city, NW Iran, and is one of the most important mica mines in Iran. The deposit is located at 45°02′14″ E longitude and 38°04′21″ N latitude (Figure 1). The study area is situated in the northern part of the Sanandaj-Sirjan Zone (SaSZ) (Figure 1a). Earlier studies [27,28] have proposed that NW Iran consists of a core of Ediacaran (Late Neoproterozoic–Early Cambrian: 500–600 Ma) igneous and metamorphic rocks [29–31]. The Ediacaran fragments are stitched together by Devonian–Carboniferous granite [32–34], Cretaceous complex [35,36], and Paleogene granite and volcanic rocks [37] (Figure 1b) and is mainly covered by Cenozoic sedimentary rocks. The Ghareh Bagh pegmatite was generated at the boundary of Carboniferous granite and Ediacaran fragments (Figure 2).

The Ghushchi complex, which was injected into the Ediacaran metamorphic rocks, is the main igneous body in our study area (Figure 2). The Ghushchi complex covers an area with 300 km$^2$. It consists of gabbro and granite and is located between Permian–Triassic sediment in SW and Urmia lake deposit in the NE. The Ghushchi complex is divided into two phases: (1) alkali feldspar granite with extensive exposure and (2) syeno-granite, which occurred in the gabbro and alkali feldspar granite. Ghushchi alkaline granites with pink color are widely exposed in the study area and was cut by some aplitic dikes. Petrographic studies in Ghushchi alkaline granites with medium grain

size of the minerals consist of orthoclase, microcline, quartz, and plagioclase. The Ghushchi complex has been well documented mineralogically, geochemically, and isotopically by Moghadam et al. [27]. The crystallization age of alkaline granites and related gabbroic bodies is 320 Ma by U-Pb zircon dating [33]. Furthermore, zircon U-Pb ages of the host Ediacaran metamorphic rocks is 570 Ma [33]. In the south part, Permian limestone (Elika formation) is exposed. Additionally, Triassic limestone and Miocene shale, sandstone, and limestone are widely distributed in this area.

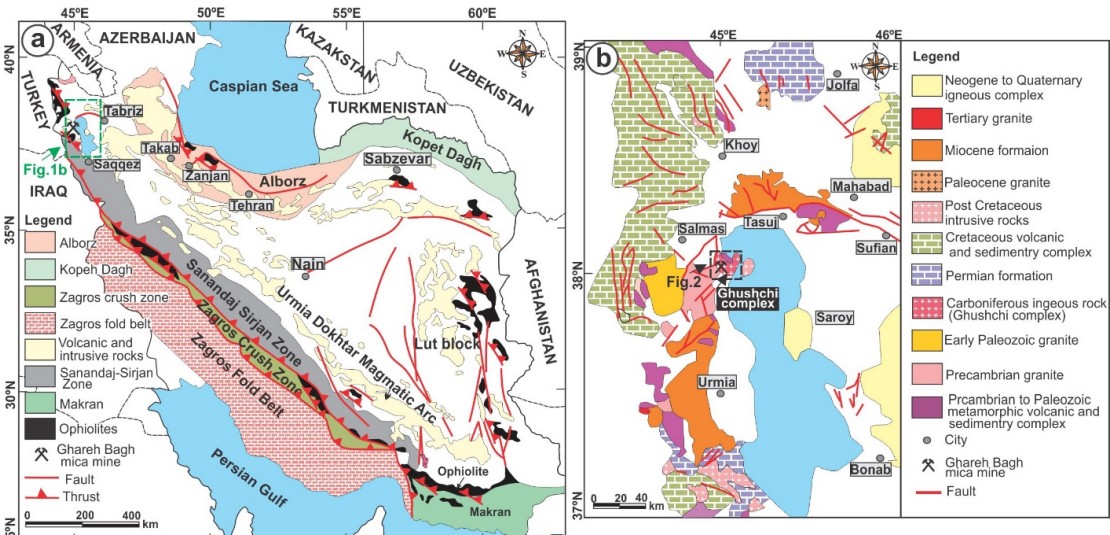

**Figure 1.** (**a**) The structural geological map of Iran [38]. The Ghareh Bagh mica mine is located in the northern Sanandaj-Sirjan Zone (SaSZ). (**b**) Simplified geological map of NW Iran [39,40].

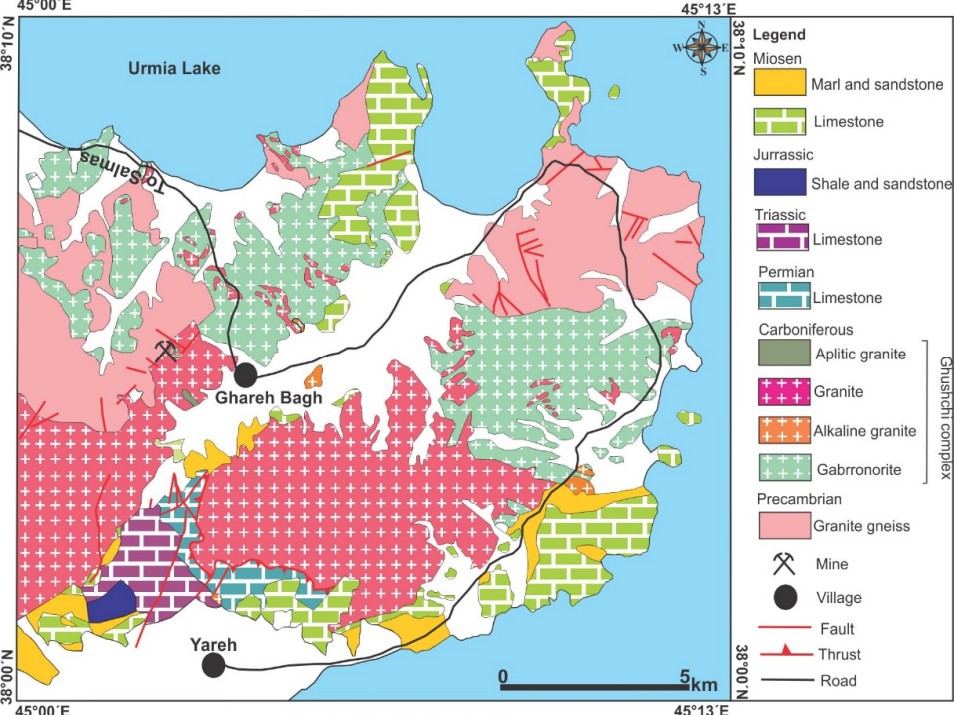

**Figure 2.** Geological map of the study area [41]. The Ghareh Bagh mine is situated at the contact of Ghushchi alkali granite and Ediacaran metamorphic rocks.

Ghareh Bagh deposit is approximately 200 m long and 50–100 m wide with a pseudo-layer and vein shapes, that extended in the NE–SW trend inside the granite body near the contact zone (Figure 3a). The granite body near the mica vein is highly brecciated (Figure 3b) and is covered by the calcic and sodic alteration. Furthermore, in the contact of the mica veins, the granitic rocks are strongly altered, and altered minerals such as clay minerals and iron oxides are developed on the western side of the mica deposit. These lines of evidence show the hydrothermal process and fluid pressure fractured the granite bodies and hydrothermal fluids infilled the cracks as matrix dominated by chlorite, biotite, actinolite, epidote, K–felspar and calcite (Figure 3c,d). Phlogopite appears as pseudo-layers and in the veins (Figure 3e). Phlogopite bands reach one meter in thickness. Epidote appears in the brecciation zone with phlogopite (Figure 3f). Apatite crystal is too rare, and some large crystals appear. The larger ones are distinguished in the mica patches (Figure 3g). Apatite occurs in green and olive green color, has a euhedral shape, and sometimes reach to 12 cm length. Calcite appears with apatite and phlogopite in pink color (Figure 3h) and has a large size around 10 cm.

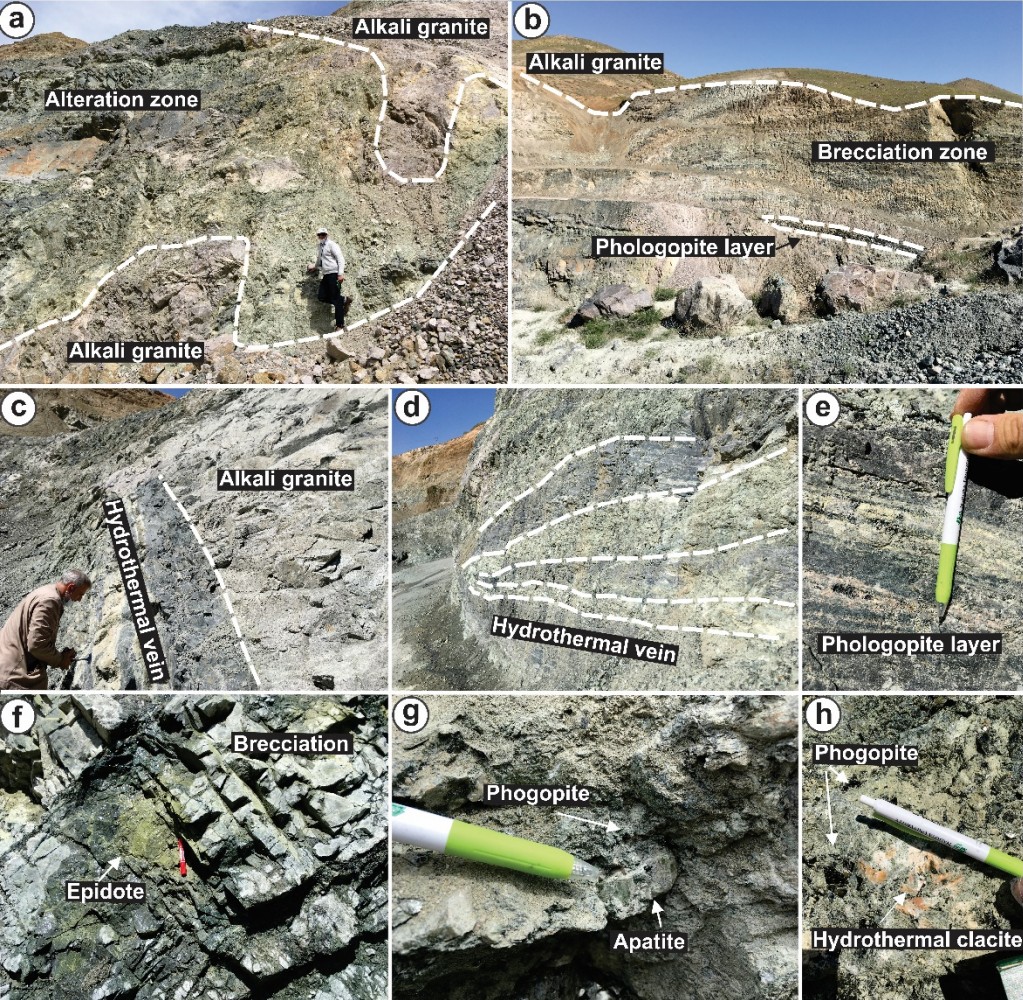

**Figure 3.** Photographs of field observation in the Ghareh Bagh mica mine. (**a**,**b**) A Ghareh Bagh mica mine pit. Photos show Ghareh Bagh mine located in alkali granite in the breccia zone with the high alteration. Mine is as layer trended in NW-SE. (**c**,**d**) Hydrothermal vein including the matrix of chlorite, biotite, actinolite, K-feldspar, calcite and quartz. (**e**) Phlogopite layer with NW-SE. (**f**–**h**) Coarse-grained minerals in parageneses of phlogopite in breccia are epidote, apatite and calcite.

### 3. Ore Deposit Mineralogy

The paragenetic sequence of mineralization and hydrothermal alteration was indicated based on microscopic studies and field observations and is shown in Figure 4. The main mineral is phlogopite which is mining now. Phlogopite–apatite mineralization is associated with the early magmatic-hydrothermal alteration in the brassica zone. The emplacement of the orebody was controlled by brecciation; according to the field observation, the mineralization zone is considered with the large crystals of phlogopite, apatite, epidote, and calcite in the brecciation zone. This assemblage is associated with an early stage of brecciation and mineralization. There is a temporal and spatial association between alteration and breccia. The calcic–sodic alteration assemblage are including chlorite, biotite, actinolite, K-feldspar, albite, calcite and quartz. The post-brecciation alteration assemblage is detected in the inner part of the orebody and overprinted on the early stage deposits and then epidote, K-feldspar, chlorite and quartz minerals were crystallized.

| Minerals | High temprature alteration Calcic- sodic alteration | Low temperature alteration |
|---|---|---|
| Phologopite | | |
| Apatite | | |
| Calcite | | |
| Epidote | | |
| Albite | | |
| K- feldspar | | |
| Actinolite | | |
| Cholrite | | |
| Quartz | | |
| Brecciation | | |

**Figure 4.** Mineral paragenesis of Ghareh Bagh mica mine.

### 4. Analytical Techniques

In this research, apatite crystals were collected from biotite bearing granite in the Ghareh Bagh mica mine for which petrological and geochemical surveys have already been performed. Apatite grains with 4 to 12 cm in diameter were separated in Ghareh Bagh mine (Figure 5), and all apatite crystal have pyramidal shapes in green color (Figure 5). In total, eight apatite samples were selected for rare earth element compositions (REEs) analysis and Sr isotope ratios. The REEs involve the light REEs (La to Sm) and heavy REEs (Eu to Lu) with atomic number 57 to 71 and also Y with atomic number 39 [42]. The big grains were checked in detail to exclude altered parts, and micro drill was used to select the pure parts of the apatite grains in Nagoya University, Japan. Approximately 10–50 mg of each mineral sample was crushed with agate miller to avoid any contamination. The samples powders were digested in the 6 molar HCl on the hot plate, and then the dissolved solution was used for quantitative analysis of REE abundances and Sr isotope analyses.

Trace elements were measured using an inductively coupled plasma mass spectrometer (ICP-MS, Agilent 7700x) at Nagoya University, Nagoya, Japan. Analytical errors for trace elements and REEs were less than 5%. The isotope ratios of Sr were measured by VG Sector 54-30 thermal ionization mass spectrometer (TIMS) at Nagoya University. $^{87}Sr/^{86}Sr$ ratios are normalized to $^{86}Sr/^{88}Sr = 0.1194$ during the measurement.

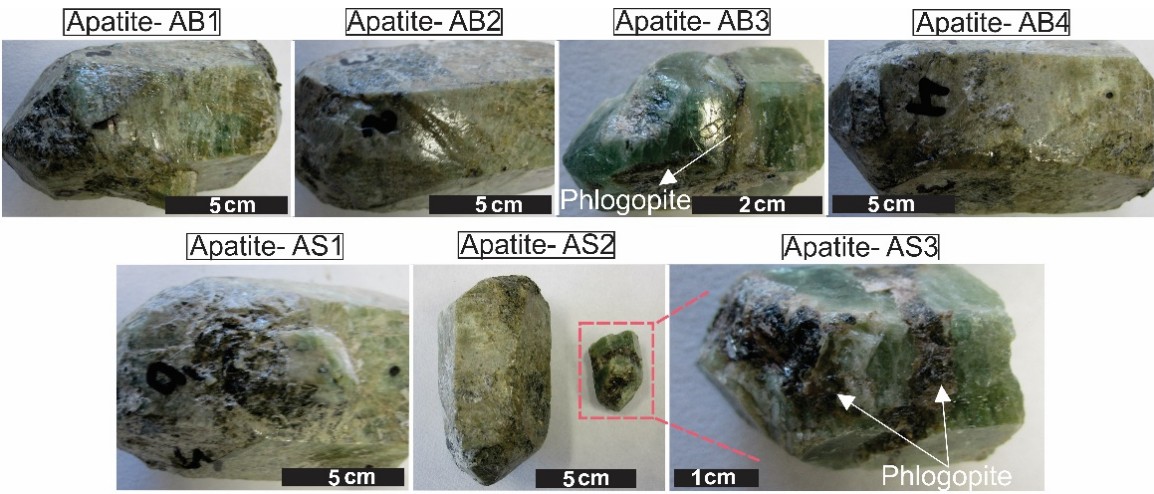

**Figure 5.** Separated apatite grains from the Ghareh Bagh mica mine. All apatite crystals have pyramidal shapes, appear in green color and grow till 12 cm. They are in the paragenesis of phlogopite.

## 5. Results

### 5.1. Apatite Chemistry

The results of the REEs concertation of the apatite grains are listed in Table 1. Total concentrations of light-REE (LREEs) in the apatite grains from the Ghareh Bagh mine vary from 3409 to 5274 ppm (average = 3979 ± 647 ppm) and the total REEs (La to Lu concentration) range from 3579 to 5619 ppm (average = 4201 ± 708 ppm). All samples have LREEs > HREEs with LREE/HREE ratios vary from 15.3 to 20.2 with an average value of 18.3. Yttrium content ranges from 236 to 497 ppm (average = 313 ± 93 ppm) (Table 1).

The chondrite-normalized REE distribution pattern for all of the apatite grains from the Ghareh Bagh mine show similar pattern and have steep decreasing trend from the LREE to HREE with $(Ce/Yb)_N$ from 19.5 to 26.2 (average = 23.8 ± 2.45) (Figure 6 and Table 1). All of the apatite grains show negative Eu anomalies with $Eu/Eu^*(Eu_N / \sqrt{Sm_N * Gd_N})$ ratios from 0.42 to 0.47 (average = 0.45 ± 0.01) (Figure 6 and Table 1).

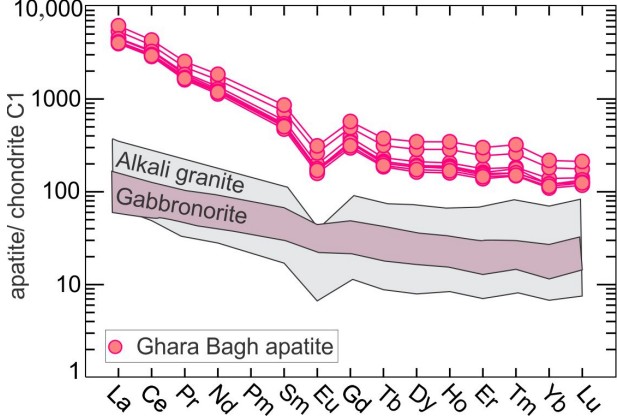

**Figure 6.** Chondrite C1-normalized rare earth element (REE) pattern of Ghareh Bagh mica mine. C1-normalizing values are from McDonough and Sun [43]. Data of the Ghushchi complex including alkali granite (grey field) and gabbronorite (purple field) from Moghadam et al. [27]. The REE normalized pattern indicates enrichment in LREE and negative Eu anomaly and shows a similar pattern to the Ghushchi Alkali granite.

**Table 1.** REE abundances of apatite crystals in the Ghareh Bagh mica deposit (in ppm).

| Sample Name. | AB1 | AB2 | AB3 | AB4 | AS1 | AS2 | AS3 | AS4 | Average |
|---|---|---|---|---|---|---|---|---|---|
| Y | 301 | 415 | 497 | 273 | 270 | 260 | 236 | 250 | 313 |
| La | 1070 | 1270 | 1458 | 982 | 1053 | 977 | 935 | 953 | 1087 |
| Ce | 1993 | 2277 | 2586 | 1824 | 1946 | 1811 | 1721 | 1759 | 1990 |
| Pr | 191 | 219 | 253 | 175 | 184 | 171 | 161 | 166 | 190 |
| Nd | 626 | 739 | 849 | 563 | 587 | 557 | 520 | 544 | 623 |
| Sm | 88.1 | 110 | 129 | 78.3 | 81.2 | 77.1 | 71.0 | 75.3 | 88.7 |
| Eu | 11.9 | 15.8 | 18.7 | 10.4 | 10.6 | 10.1 | 9.46 | 10.2 | 12.1 |
| Gd | 77.2 | 98.0 | 114 | 69.2 | 66.8 | 67.8 | 59.4 | 62.6 | 76.9 |
| Tb | 9.31 | 12.6 | 15.0 | 8.49 | 8.21 | 8.06 | 7.43 | 7.68 | 9.60 |
| Dy | 52.7 | 71.6 | 86.6 | 47.9 | 45.7 | 46.5 | 40.8 | 43.3 | 54.4 |
| Ho | 10.2 | 14.3 | 17.3 | 9.46 | 9.25 | 8.93 | 7.97 | 8.41 | 10.7 |
| Er | 28.3 | 39.2 | 47.9 | 25.8 | 25.1 | 24.3 | 22.1 | 23.2 | 29.5 |
| Tm | 3.66 | 5.16 | 6.43 | 3.54 | 3.19 | 3.20 | 2.97 | 3.01 | 3.90 |
| Yb | 22.2 | 29.0 | 34.8 | 19.2 | 19.5 | 19.4 | 17.5 | 18.3 | 22.5 |
| Lu | 2.83 | 3.52 | 4.24 | 2.68 | 2.51 | 2.53 | 2.35 | 2.49 | 2.89 |
| $\sum$REE | 4186 | 4904 | 5619 | 3819 | 4042 | 3784 | 3579 | 3676 | 4201 |
| La/Nd | 1.71 | 1.72 | 1.72 | 1.74 | 1.80 | 1.75 | 1.80 | 1.75 | 1.75 |
| Ce/Yb$_N$ | 23.6 | 20.6 | 19.5 | 24.9 | 26.2 | 24.6 | 25.8 | 25.2 | 23.8 |
| (La/Yb)$_N$ | 32.7 | 29.7 | 28.5 | 34.7 | 36.7 | 34.3 | 36.3 | 35.3 | 33.5 |
| (La/Sm)$_N$ | 7.58 | 7.22 | 7.07 | 7.83 | 8.10 | 7.91 | 8.23 | 7.90 | 7.73 |
| (La/Gd)$_N$ | 11.6 | 10.9 | 10.7 | 11.9 | 13.2 | 12.1 | 13.2 | 12.8 | 12.1 |
| (Gd/Yb)$_N$ | 2.81 | 2.73 | 2.66 | 2.92 | 2.77 | 2.83 | 2.74 | 2.76 | 2.78 |
| LREE | 3968 | 4615 | 5274 | 3622 | 3851 | 3593 | 3409 | 3497 | 3979 |
| HREE | 218 | 289 | 345 | 197 | 191 | 191 | 170 | 179 | 223 |
| HREE/LREE | 18.2 | 16.0 | 15.3 | 18.4 | 20.2 | 18.8 | 20.1 | 19.5 | 18.3 |
| Eu/Eu*[1] | 0.44 | 0.46 | 0.47 | 0.43 | 0.44 | 0.42 | 0.44 | 0.45 | 0.45 |
| Ce/Ce*[2] | 1.07 | 1.04 | 1.03 | 1.06 | 1.07 | 1.07 | 1.07 | 1.07 | 1.06 |
| Y/Y*[3] | 0.95 | 0.96 | 0.95 | 0.95 | 0.97 | 0.94 | 0.96 | 0.96 | 0.95 |

N denotes chondrite normalization. [1] $Eu/Eu^* = Eu_N / \sqrt{Sm_N \times Gd_N}$; [2] $Ce/Ce^* = Ce_N / \sqrt{La_N \times Pr_N}$; [3] $Y/Y^* = Y_N / \sqrt{Dy_N \times Ho_N}$.

## 5.2. $^{87}Sr/^{86}Sr$ Ratios

$^{87}Sr/^{86}Sr$ ratios of the eight apatite grains are presented in Table 2. The $^{87}Sr/^{86}Sr$ ratios clearly show the same values with minor variation and change from 070917 to 0.70950 with an average = 0.70929 ± 0.00011 (*n* = 8, 1SD).

**Table 2.** Sr isotope compositions in apatite of Ghareh Bagh.

| Sample | AB1 | AB2 | AB3 | AB4 | AS1 | AS2 | AS3 | AS4 |
|---|---|---|---|---|---|---|---|---|
| $^{87}Sr/^{86}Sr_{(p)}$ | 0.709182 | 0.709449 | 0.709501 | 0.709244 | 0.709250 | 0.709211 | 0.709170 | 0.709184 |
| 1 sigma error | 0.000006 | 0.000006 | 0.000006 | 0.000008 | 0.000006 | 0.000006 | 0.000006 | 0.000007 |

## 6. Discussion

### 6.1. Apatite Genesis

Apatite generally occurs as an early stage crystallizing mineral in the various type of magma [18], and reflects the early magma geochemical information [44]. Additionally, apatite precipitate appears in the hydrothermal processes. Accordingly, apatite is a useful mineral as a petrogenetic indicator and oxygen fugacity [9,14].

The distribution coefficient of REE in the silica melt for apatite is 12–20, which occupy Ca sites [45]. Therefore, REE strongly concentrates on the apatite and is incompatible with silicates and oxides [46]. REE composition in apatite reflects the composition of melt (SiO$_2$ content, REE abundant,

oxidation state) and fluids which if forms [47]. Therefore, the magma composition and the degree of magmatic differentiation and co-crystallizing minerals can control the shape of the normalized REE patterns in apatite [2]. The REE-normalized pattern of the Ghareh Bagh's apatite shows a significant LREE enrichment relative to the HREE with Eu anomalies (Figure 6). Additionally, they have a high content of REE (more than 1000 ppm). The REE pattern and high LREE contents in the apatite grains are consistent with magmatic apatite notably in the alkaline rock which is marked by slightly inclinedly REE pattern with negative Eu anomaly [6,23,48]. Meanwhile, the LREE distribution coefficient in the fluid to melt $(D_{LREE}^{f/M})$ is higher than the $D_{HREE}^{f/M}$ in apatite [49–51], and also, in the melt $D_{REE}^{ap/M}$ increases with increasing $SiO_2$ content [52]. So, the early magmatic-hydrothermal fluid could enrich LREE more than the MREE and HREE in Ghareh Bagh's apatite as well. The transition from magmatic to hydrothermal could be distinguished by behavior of Y, which negative Y-anomaly could be associated by behavior of Y as a pseudo-lanthanide partitioning away due the other REE in F-rich fluids [53]. The slight negative Y (Y/Y*)(average = 0.96 ± 0.01) is observed in the Ghareh Bagh's apatite. The distribution of the apatite in brecciation zone with phlogopite, epidote and exposure of extensive alteration zone show fluids activities was too high during the generation of ore bearing minerals in the Ghareh Bagh area. In addition, the REE normalized pattern does not match with hydrothermal apatite at low temperature. The typical REE distribution of low temperature apatite is diagnostic with low Eu contents, depletion in the LREE, and enrichment in the MREE [15,54,55]. It seems apatite came up during the early magmatic-hydrothermal fluids with the higher temperature. The phlogopite forming temperature is estimated according to the thermometry apatite–phlogopite around 600 °C [56], which confirm the high temperature magmatic-hydrothermal sources for apatite grains [52].

By consideration of REE pattern in the Ghareh Bagh apatite, negative Eu anomaly (average = 0.45 ± 3.31) and high $(Ce/Yb)_N$ ratios (average = 23.8± 9.36) infer the hydrothermal apatite grains were formed in the early exsolved fluid at the higher temperature directly from the feldspar fractionated felsic melts [16]. One application of the Eu anomaly is that it could be a good indicator of the timing of fluid exsolution from the magma, for example, early or late stage exsolution. The negative Eu anomaly shows fluid exsolved in the early stage in the depth and high positive Eu anomaly indicates late-stage exsolved fluid during the ascending to the shallow depth [52]. This is because $Eu^{2+}/Eu^{3+}$ control the Eu anomaly, $D_{Eu^{+2}}^{f/M}$ is unlike the $D_{REE^{+3}}^{f/M}$ and it increases with decreasing the pressure⁻. Hence, Eu anomaly is positive in the late stage exsolved fluids at the lower depth [50,52,57]. Accordingly, REE content decreases in fluids that exsolved in shallow depth [57].

The Gharah Bagh's apatite shows a negative Eu anomaly (Figure 6) which confirms that the fluid might exsolve in the high depth. Adlakha et al. [52] used $(La/Sm)_N$ versus Eu/Eu*, diagram to infer the sources of the apatite. All of the Ghareh Bagh apatite are plotted in the boundary of the felsic melt/hydrothermal fluid during fractionation crystallization (Figure 7). Additionally, $D_{LREE}^{f/M}$ is generally less than one (<1) which might indicate the high salinity of the associated fluids [52]. Therefore, REE-enriched apatite in Ghareh Bagh seems to be formed by a very saline hydrothermal-magmatic fluid.

Due to the location of the Ghareh Bagh deposit at the contact of the Gushchi granite with granite gneiss of the Precambrian complex, there are some possibilities for the reaction of the granite with host rocks to produce the mica deposit. Apatite from migmatites and metamorphic rocks has low ratio of $(Ce/Yb)_N$ (≤1) with approximately flat to depleted LREE pattern [58,59]. Our report in this study result does not confirm the genesis of the apatite during the metamorphic reaction, because the Ghareh Bagh apatite has higher ratios of $(Ce/Yb)_N$ (19.5 to 26.2). Metamorphic apatite are extremely heterogeneous and have lower $\sum REE$ [8,60]. However, the Ghareh Bagh apatite is too enriched in REE (3579 to 5619 ppm).

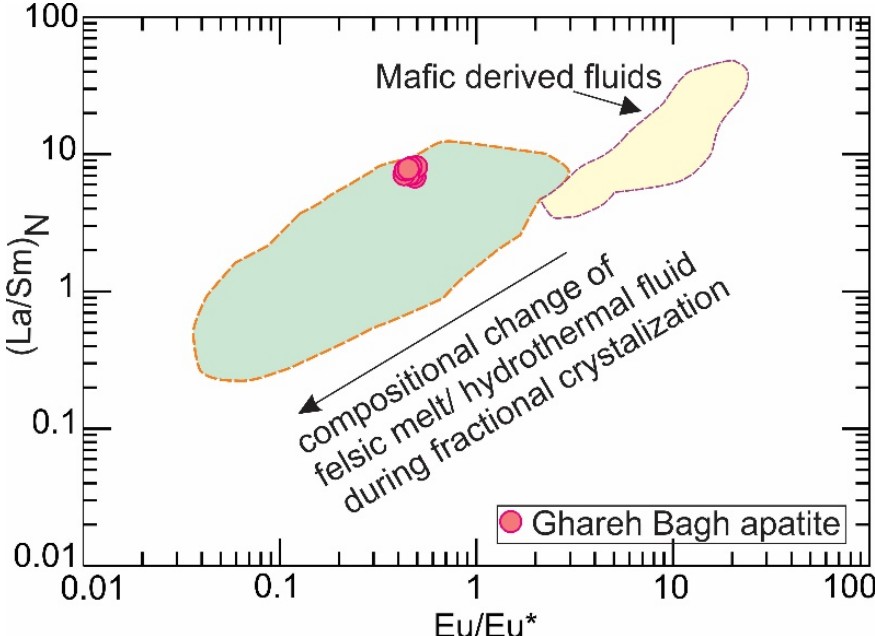

**Figure 7.** Biplot of La/Sm vs. Eu/Eu*. Fields of hydrothermal derived felsic and mafic melt from Adlakha et al. [52]. The Ghareh Bagh's apatite are plotted in a hydrothermal-derived felsic magma.

*6.2. Apatite and Its Relation to the Host Granite*

Apatite geochemistry could be used as a fingerprint of magma source, post crystallization hydrothermal event, and oxide state fluid and melt [48]. Belousva et al. [2] proposed that some elements like REEs are the key index to determine the compositional apatite-bearing rocks. Therefore, using the Y versus Eu/Eu* discrimination diagram [2], all given apatite with the Y content ranging from 236 to 497 ppm and moderately negative Eu anomalies fall in the granite field (Figure 8a). The mineral–melt distribution coefficients can control the content of REEs in the apatite. Therefore, the REE distribution pattern in the apatite is similar to the bulk REE composition of source magma [61]. The comparison of the chondrite normalized REE, distribution pattern in apatite and alkali granite and gabbronorite of Ghushchi batholith indicate that the REE pattern of apatite and alkali granite are very similar (strong LREE/HREE and Eu anomaly) (Figure 6). These similar patterns show that alkali granite intrusion and apatite grains are generated from the same magma source.

In addition, the $(La + Ce + Pr)/\sum REE(\%)$ ratios would be much useful to understanding of the sources of apatite [58]. The alkaline signature is clarified by using La/Nd ratios versus the $(La + Ce + Pr)/\sum REE(\%)$ diagram, which is a main discrimination diagram for the sources of apatite grains from the different rocks [58]. The Ghareh Bagh apatite samples with La/Nd ratios from 1.71 to 1.80 (mean = 1.75) and $(La + Ce + Pr)/\sum REE(\%)$ from 76.5 to 78.8 (mean = 77.9) (Table 1) plot in the alkaline rocks field (Figure 8b). This shows clear consistence with the REEs pattern which we have discussed in the previous section. The REE normalized pattern for the Ghareh Bagh apatite is characterized by inclined REE pattern with LREE enrichment and negative Eu anomaly which seems to show affinity to alkaline magmatism in extensional regime [48]. Moreover, the euhedral form of the apatite crystal with high values of the REE is diagnostic of apatite from alkaline magmatism [62] which are highlighted for the Ghareh Bagh samples.

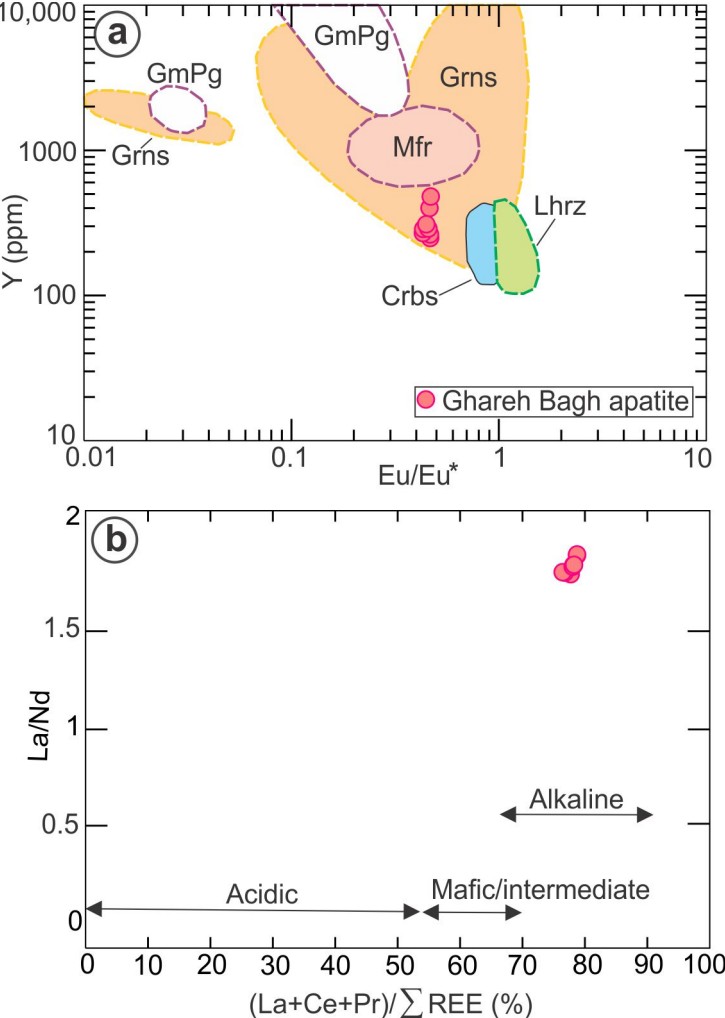

**Figure 8.** (**a**) Y vs. Eu/Eu* genetic diagram [2], showing composition from all of the studies rock. The apatite shows that they are derived from granitoid. (**b**) Apatite samples plot on the La/Nd vs. (La + Ce + Pr)/$\sum REE$(%) classification diagram of Fleischer and Altschuler [58]. Data indicate alkaline affinity. Abbreviations; Lhrz = lherzolites, Crbs = Carbonatites, Prx = Pyroxenites, Lrvk = larvikite, Dlr = dolerites, Grns = granitoids, GrnPg = felsic pegmatite and Mfr = mafic rocks.

Apatite is more abundance minerals in the A-type granite which has been divided into two groups $A_1$ and $A_2$ by Eby [63]. $A_1$-type is related to oceanic island basalt (OIB) sources and $A_2$–type is produced from the partial melting of juvenile continental material. These subgroups are generated in an extensional regime with different sources [63]. REE normalized patterns in both $A_1$ and $A_2$ groups are similar and are marked by highly LREE enrichment and negative Eu–anomaly [48] (Figure 6). Nevertheless, because of the different sources, apatite in $A_2$–type shows a higher $(La/Yb)_N$ ratio and negative correlation between the $(La/Gd)_N$ and $(Gd/Yb)_N$ ratio [48]. In the comparison of the Ghareh Bagh´s apatite with $A_1$ and $A_2$ apatite field in $(La/Yb)_N$ vs. $(Gd/Yb)_N$ and $(La/Gd)_N$ vs. $(Gd/Yb)_N$ [48], samples plot in $A_2$ group (Figure 9). Moghadam et al. [33] show $A_2$–type affinity and lower continental crust source for the Ghoushchi alkali granite, which is more consistent with our findings in this research based on the apatite REE pattern. According to Jiang et al. [48] apatite from $A_1$ and $A_2$ granite indicate the similar content of major and trace element but apatite in $A_1$ type enrich in Cl and $A_2$ type is F-rich. So, Ghareh Bagh apatite seem to be F-rich and F could liberate from breakdown of F-bearing mineral such as phengite and lawsonite in deep depths [48], which is consist with $A_2$ Ghushchi granite.

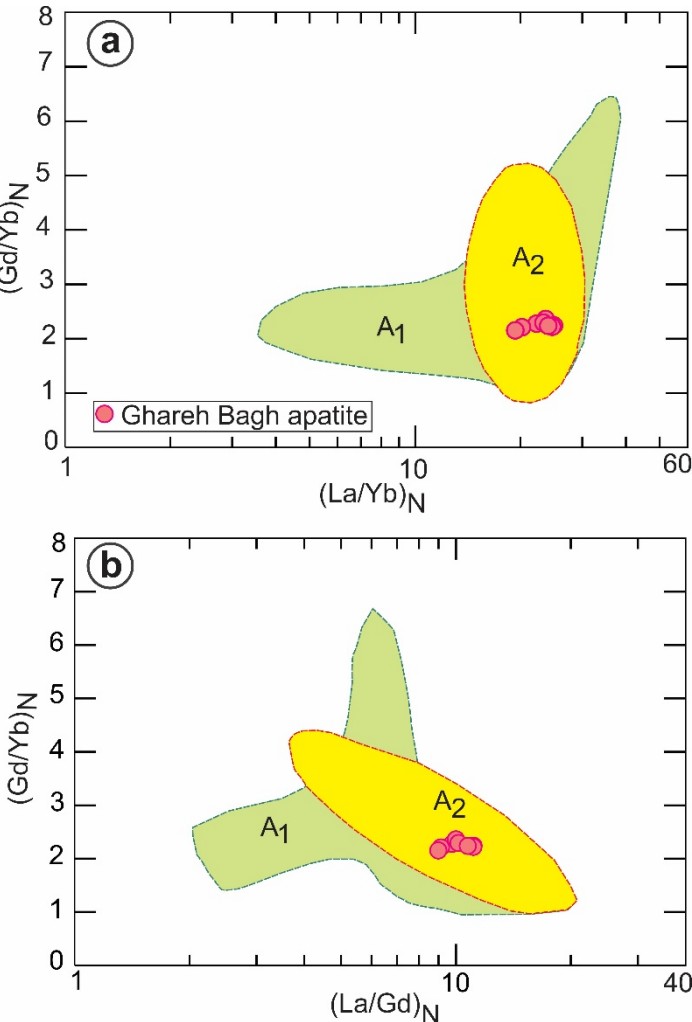

**Figure 9.** (**a**,**b**) Biplots of $(Gd/Yb)_N$ vs. $(La/Yd)_N$ and $(Gd/Yb)_N$ vs. $(La/Gd)_N$ for A-type granite. Data of apatite from $A_1$ (greenfield) and $A_2$ (yellow field) from Jiang et al. [48]. Apatite in Ghareh Bagh plot in $A_2$ filed.

*6.3. Eu Contents in the Apatite and Oxygen Fugacity*

Feldspars are the main host for Eu in its structure. Therefore, crystallization and segregation of these minerals play a key role in the content of the Eu in the residual melts and exsolved fluids [6]. The negative Eu anomalies (Eu/Eu*) in apatite (normalized REE patterns) are mostly controlled by feldspar crystallization [6,64]. Negative Eu anomaly in hydrothermal apatite reflects the depletion of Eu in the fluids [52] because most of Eu is accompanied in the feldspar structures. Therefore, Eu negative anomaly in the Ghareh Bagh apatite is probably related to the plagioclase crystallization before the apatite crystallization. Eu/Eu* in apatite grains are related to Eu/Eu* in the melt, $Eu^{2+}/Eu^{3+}$ ratio, and the redox condition of the host magma [3,65,66]. The redox condition can control the specious of volatiles and ligand and plays a significant role in magmatic-hydrothermal systems. Apatite favorably incorporates $Eu^{3+}$ rather than $Eu^{2+}$, and the most important factor for controlling the two Eu is oxygen fugacity of magma [14]. To determine the redox condition of magma, the ratios of Eu/Eu* and Ce/Ce* in apatite is a good indicator [14], because the oxygen fugacity controls the ratio of $Eu^{2+}/Eu^{3+}$ and $Ce^{3+}/Ce^{4+}$. In the low fugacity, there is low $Eu^{3+}$ and high $Ce^{3+}$ and vice versa in high fugacity [9]. On Eu/Eu* vs. Ce/Ce* plot, all the apatite plot is in moderately oxidized magma (Figure 10).

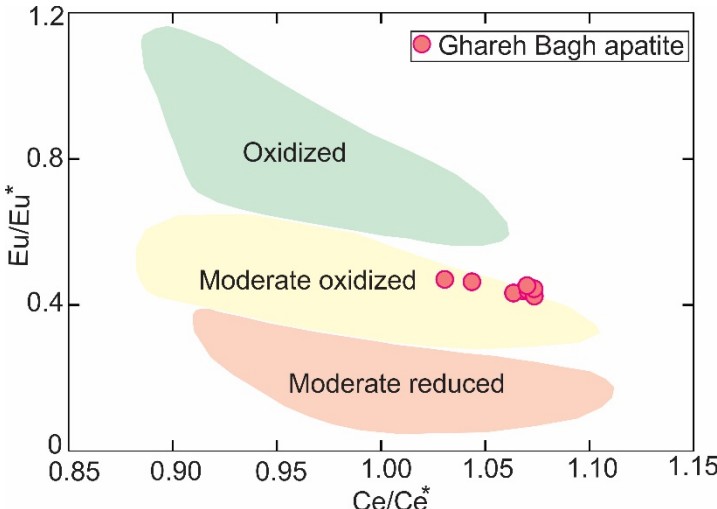

**Figure 10.** Eu anomalies (Eu/Eu*) vs. Ce anomalies for the apatite samples. Fields from Cao et al. [14]. Ghareh Bagh apatite plot in the moderate oxidized environment.

### 6.4. $^{87}Sr/^{86}Sr$ Ratios in the Apatite Granis

Because of too low Rb and high Sr contents in apatite grains, the Rb/Sr ratio is extremely low and the radioactive decay of $^{87}Rb$ is insignificant [18]. Therefore, apatite is a good candidate to preserve the initial $^{87}Sr/^{86}Sr$ ratio of the source magma [21,22,67].

The Ghareh Bagh apatite grains show the high ratios of $^{87}Sr/^{86}Sr$ (0.709170 to 0.709501). No significant variation in the $^{87}Sr/^{86}Sr$ ratios show the separated apatite grains have homogeneous composition. Generally, rapid cooling and absence of thermal distribution cause the preservation of initial $^{87}Sr/^{86}Sr$ in the apatite [21]. High $^{87}Sr/^{86}Sr$ ratios could be related to magmatic sources from lower crustal or highly metasomatic mantle sources [6,8,68] and is consistent with lower continental crust and/or subcontinental mantle source for Ghushchi intrusive. This conclusion is consistent with the bulk rock $\mathcal{E}Nd$ (+1.3 to +3.4) and zircon $\mathcal{E}Hf$ isotopic signatures (+1.7 to +6.2) in the late Paleozoic [33]. In addition, sometimes, Rb-rich minerals such as monazite as inclusion in the apatite increase $^{87}Sr/^{86}Sr$ ratio [67]. Furthermore, apatite composition in moderate to high temperature could be significantly affected with fluids, and this causes the open Sr system in apatite [4,68,69]. As we discussed above, apatite REE composition in the Ghareh Bagh mica mine was formed from the early magmatic-hydrothermal source of Ghushchi alkali granite at high temperature, so that the Sr system might have been opened.

### 6.5. Mica–Apatite Deposit Mineralization

The Ghareh Bagh phlogopite- apatite mine have brecciated texture with high-temperature calcic-sodic alteration. Based on the chondrite-normalized REE signatures, including moderate negative Eu and higher ratios of the LREE/HREE, mineral paragenesis, brecciated texture, Ghareh Bagh apatite may have been directly precipitated from the magma in the early hydrothermal stage at the high temperature from Ghuchshi alkaline granite, and this is consistent with immiscible melt hypothesis [16]. A petrogenetic relationship between magma and mineralization is supported by field relations and apatite chemistry.

The igneous apatite could be transported up by magmatic-hydrothermal fluid and comes up as fluid suspension [70,71]. During the cooling of alkaline magma, a high volume of alkaline-rich volatile increases, and therefore, volatiles cause overpressure and brecciated the surrounding rocks [70]. The alkaline fluid was responsible for alteration mineral paragenesis such as biotite, K-feldspar, albite, actinolite and epidote in the brecciation zone. The composition of the fluids is expectedly dominated by alkaline cations such as $Na^+$, $K^+$ and $Ca^{2+}$ [72–74]. The brine fluid could scavenge

LREE from silicate melt [75]. During ascending to the fluid suspension, apatite continues to grow from magmatic-hydrothermal fluid [75], and destabilized precipitated by rapid transport in the hydraulic fracture in breccia [76]. It is well documented that F values in the intrusive rocks increase with increasing alkalinity [77] and therefore, high F content is characterized in magma with alkaline affinity [73]. F-bearing system consists F-rich apatite associated with $A_2$–type signature of Ghushchi granite biotite is the main OH–F bearing phase [78]. In high-temperature F-rich system, biotite has high-Mg content [79] and high Mg/Fe ratio [80]. So, in the Ghareh Bagh mine, phlogopite could be precipitated in the breccia with epidote, chlorite K-feldspar, and albite related to the composition of brine hydrothermal magmatic fluid. The thermometry of the phlogopite–apatite couple infers the high temperature (600 °C; [56]). Phlogopite and apatite are the main halogen and REE stores in the Ghareh Bagh deposit. Therefore, geochemical investigation of biotite and apatite and their relationship help us to interpret and support the genetic model of the Ghareh Bagh mine.

Radiogenic isotope is a valuable tool to determine the source of magmatic-hydrothermal agents. Our hypothesis is in agreement with the phlogopite K/Ar dating of 319.4 ± 8.2 Ma for the Ghareh Bagh mine [56]. This confirms plutonism and extensional regime by Moghadam et al. [33] which showed gabbronorite and alkali granite in Ghushchi complex are synchronously at 320 Ma ago, and phlogopite in Ghareh Bagh are roughly synchronous. The phlogopite–apatite veins have been precipitated from the early magmatic-hydrothermal brine fluid in the extensional tectonic regime in the brecciation zones. Alkali granite and gabbronorite have similar $\varepsilon Nd(t) =$(+1.3 to +3.4 and 0.1 to 4.4; respectively) and $\varepsilon Hf(t) =$ (+1.7 to 6.2 and +0.94 to +6.5; respectively) and gabbronorite are similar to OIB in geochemistry [33]. Our suggested genetic model is summarized in Figure 11. In bases on our schematic model, early stage magmatic-hydrothermal fluid with high temperature was response for the hydraulic fracture and brecciations of the host rocks and follows, biotite (phlogopite), epidote, chlorite K-feldspar and albite were precipitated in that fractures. The retrograde hydrothermal fluid with low contents of Na, Ca, and REE was overprinted the early stage mineralization in this area.

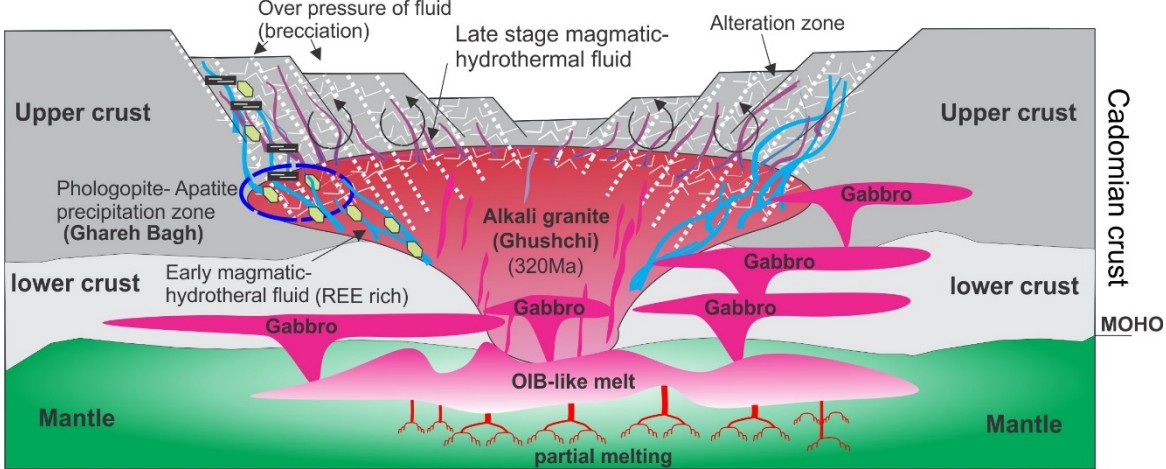

**Figure 11.** A conceptual genetic model of phlogopite and apatite formation in the Ghareh Bagh mica mine with breccia, alteration and tectono-genetic relations with Ghushchi alkaline magma and gabbronorite based on apatite composition.

## 7. Conclusions

We conclude that Ghareh Bagh's apatite large crystals were originated from the alkaline magmatic-hydrothermal fluid at high temperature. During the degassing of magma, apatite could be swept up and suspended in the fluid. This fluid is buoyant, so ascend from the source of magma. Destabilizing of apatite happens in the brecciated zone because of the rapid transport. The composition

of the magmatic-hydrothermal fluid is dominated by $Na^+$, $K^+$ and $Ca^{2+}$. It is common in alkaline fluid and causes the precipitation of phlogopite, epidote, chlorite, K-feldspar and albite in the brecciated zone. The high $^{87}Sr/^{86}Sr$ ratios in the apatite crystals infer the lower crust source of magma and/or open Sr system at the high temperature. Synchronous reported K-Ar ages of the phlogopite grains and host granites to show that the vein mineralization and host rocks crystallization almost occurred at the same time. REE pattern in apatite could be an important fingerprint of melt and fluid during the formation and gives us much information about both mineralization and magma sources.

**Author Contributions:** H.A. and Y.A. performed all analytical work. Data processing was done by N.D. under the guidance of H.A. and M.T., M.H. contributed samples and H.A. has done the fieldwork. N.D. and H.A. wrote the paper, assisted by all other authors. All authors have read and agreed to the published version of the manuscript.

**Funding:** This research was funded by the Japan Society for Promotion of the Science (JSPS) KAKENHI grant number 17H01671.

**Acknowledgments:** The authors would like to thank K. Abasszadeh for fieldwork assistance. This version much benefited from both anonymous reviewers' and assistant editor comments.

**Conflicts of Interest:** The authors declare no conflict of interest.

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
