# Peer review of "Rare Earth Elements and Sr Isotope Ratios of Large Apatite Crystals in Ghareh Bagh Mica Mine, NW Iran: Tracing for Petrogenesis and Mineralization"

_minerals, doi:10.3390/min10090833_

Round 1
Reviewer 1 Report
Dear Authors,
Thank you for your submission to the Minerals. I went through your manuscript a few times. You have done great sampling and analytical work, but when it comes to presenting your work, lots of detail is missing and the majority of the discussion is based on assumptions from previous works. You also do not present any evidence to support your conclusions and do not explain your petrogenetic model. Please see the attached file for your reference.

Author Response
Response to the comments
We are really appreciated for your clear editing; meanwhile all of your comments are added to the text. Please check the new version. All change is marked by track change for your consideration.
Also, we made some corrections and add explanations about the petrographic sequence and genetic model. You can check them out in part “ore deposit mineralogy” in lines 149- 160 and “Mica- apatite deposit mineralization” in line 403-449.
All the manuscript is edited grammatically base on your suggestion phrase and verbs.
The suggested reference such as Azadbakht et al., (2018) is added.
We put some petrology information on Ghushchi granite in lines 119- 122. Because the Ghushchi complex has been well- documented mineralogically, geochemically and isotopically by Moghadam et al 2015.
Some information about the two-generation alteration in Ghareh Bagh mica mine is added in lines 150- 160 and we show the mineral sequence in the figure. 4.
The standard deviation is added for all average content in the text.
About the oxygen fugacity, unfortunately, Mn content in apatite has not been detected. Meanwhile, plagioclase crystallization could affect the Eu/Eu* ratio, so we use Eu/Eu* vs. Ce/ Ce* to get reliable information about the oxygen fugacity.
The petrogenetic model is well described in line 404- 449. we removed some part which no evidence and discuss about the property of fluids and how biotite and phlogopite is precipitation. We used the result which published to more clarify our suggested model.
The alkaline property of the fluid is clarified by the mineral alteration sequence which is mentioned in line 415 and high temperature consists of REE pattern in apatite and the published thermometry of phlogopite- apatite which is mention in line 430-431.
Please see our revised version and we hope you find it suitable for publishing in this journal.
Sincerely,
Authors

Reviewer 2 Report
I found this article very interesting. However (in addition to form-related improvements, noted in the .PDf file), I think the subject was not well conducted for the following reasons:
- lack of complete apatite analysis. What type of apatite? How much Na, Si?
Considering the discussion in the lines 329-331, the authors indicate that Ghareh Bagh apatite is a chloroapatite. If this is correct, chloroapatite is not the most common apatite in A-type granites (See the cited Jiang et al. 2018). At least, the authors should clarify this subject and present, if possible, F and Cl analysis.
- The relationship between apatite and phlogopite is not well established. If both are hydrothermal and formed by fluids derived from granite, how can one explain the presence of phlogopite instead of iron rich biotite? Besides the low mobility of Mg, how to explain such Mg enrichment in A-type granite derived fluids?
- Moghadam et al.2015 present an interesting proposal that Ghushchi Alkali granite and gabbronorite are co-genetic, but the authors do not consider this proposal on their discussions and how this relationship could influence the origin of the Ghareh Bagh apatite.

Author Response
Reviewer #2
- lack of complete apatite analysis. What type of apatite? How much Na, Si?
Considering the discussion in the lines 329-331, the authors indicate that Ghareh Bagh apatite is a chloroapatite. If this is correct, chloroapatite is not the most common apatite in A-type granites (See the cited Jiang et al. 2018). At least, the authors should clarify this subject and present, if possible, F and Cl analysis.
Response: We are completely agreed with this comment. Generally, apatite is ubiquitous mineral in A-type granites. The Ghushchi complex well document geochemically by Moghadam et al., (2015). Based on this article the source of the Ghushchi granite is A2- type and its source is partial melting of crust. Jiang et al. (2018) documented apatite from A1 and A2 granite indicate the similar content of major and trace element but differ in their F and Cl concentrations. Apatite related to A1- type is characterized by high Cl concentration and it is the result of breakdown of amphibole chlorite and/ or serpentine decomposition during partial melting of subducted oceanic crust which cause the liberation of high concentration of Cl. In contract, apatite in A2 type granite source rock enrich in F which is released of breakdown of phengite and lawsonite in deeper depths in subducted oceanic crust. So, Ghareh Bagh apatite seem to be F- rich. So, we modified the fluid description and explanation of the magma magmatism volatile, please check them out in lines 357-361.
- The relationship between apatite and phlogopite is not well established. If both are hydrothermal and formed by fluids derived from granite, how can one explain the presence of phlogopite instead of iron rich biotite? Besides the low mobility of Mg, how to explain such Mg enrichment in A-type granite derived fluids?
Response: We discuss the phlogopite are formed by high temperature hydrothermal fluid. It is well known that F contents in igneous rocks increase with increasing alkalinity. therefore, high F content is characterized in magma with alkaline affinity. F- bearing system consist F-rich apatite associated to A2 type signature of Ghushchi granite biotite is the main OH- F bearing phase. In high temperature F rich system, biotite has high Mg content and high Mg/Fe ratio. So, in Ghareh Bagh mine phlogopite could precipitated in the breccia with epidote, chlorite K-feldspar, and albite related to the composition of brine hydrothermal magmatic fluid. The thermometry of phlogopite- apatite pair clarify high temperature (600°C). Please check the manuscript lines 412- 436.
- Moghadam et al.2015 present an interesting proposal that Ghushchi Alkali granite and gabbronorite are co-genetic, but the authors do not consider this proposal on their discussions and how this relationship could influence the origin of the Ghareh Bagh apatite.
Response: Moghadam et a., (2015) well documented Ghushchi complex including alkali granite and gabbronorite. Based on U-Pb dating in zircon, gabbronorite and alkali granite emplaced in the same time in 320 Ma. They have similar (+1.3 to +3.4 and 0.1 to 4.4; respectively) and (+1.7 to 6.2 and +0.94 to +6.5; respectively). They have similar geochemically and gabbronorite show the OIB affinity. So, we modified the genetic model based on relationship between granite and gabbro. Also, we have showed the genesis of the phlogopite- apatite mineralization and trying to draw our model base on by using geochemistry of apatite. Please check the manuscript in lines 440-445 and figure. 11.
The magma petrogenesis and geodynamic clarify of magmatism in the 340-310 Ma in the NW Iran and the previous researchers made clearly discussion on the magmatic evolution in here based on the Zirocn U-Pb ages, whole rocks compositions and Sr-Nd isotope ratios(Saccani et al 2013, Moghadam et al., 2015; Azizi et al., 2017 ) and all of them have reported the same ages for both gabbroic to granite rocks in an extensional tectonic regime. In addition, these researchers believe to the A- type granite in here and also Azizi et al (2017) have shown the migration from the A1 to A2 type granite in NW Iran in this period. Furthermore, our new submitted work with radiometric ages in 330 Ma in western Saqqez (southern Ghoshchi) is almost consistent with previous published works. Nevertheless, our aim in this work is the late stage of magma crystallization and precipitation of the apatite in the F-rich fluids.
We are looking to hear good news from you.
Sincerely;
Authors

Round 2
Reviewer 1 Report
Dear Authors,
Thank you for your hard work in editing your manuscript. I can see great improvement! There are still a few places that need extra work, and I believe that the manuscript would benefit from an English editing.
With kind regards,
Reviewer 1

Author Response
Dear Reviews,
We are appreciated for your comment. We improved English and based on our experiences it is Ok. If you think it needs more we will apply for some company.
Please see both version mark and clean version
Best regards,
Hossein Azizi
Reviewer 2 Report
The authors implemented the changes suggested in the first review. I am satisfied with the answers and clarifications provided in the manuscript and cover letter. However the changes in the text were substantial and although I am not an expert in the English language, I noticed problems in some sentences that seem to be garbled. So it seems to me that an editing of English language and style is required
Author Response
Dear Reviews,
We are appreciated for your comment. We improved English and based on our experiences it is Ok. If you think it needs more we will apply for some company
Please see both version mark and clean version.
Best
Hossein Azizi